# Kinesin-1 regulates dendrite microtubule polarity in *Caenorhabditis elegans*

**Jing Yan[1], Dan L Chao[1], Shiori Toba[2], Kotaro Koyasako[3,4], Takuo Yasunaga[3,4,5], Shinji Hirotsune[2], Kang Shen[1]\***

[1]Department of Biology, Howard Hughes Medical Institute, Stanford University, Stanford, United States; [2]Department of Genetic Disease Research, Graduate School of Medicine, Osaka City University, Osaka, Japan; [3]Department of Bioscience and Bioinformatics, Faculty of Computer Science and Systems Engineering, Kyushu Institute of Technology, Fukuoka, Japan; [4]JST-SENTAN, Japan Science and Technology Agency, Kawaguchi, Japan; [5]JST-CREST, Japan Science and Technology Agency, Kawaguchi, Japan

**Abstract** In neurons, microtubules (MTs) span the length of both axons and dendrites, and the molecular motors use these intracellular 'highways' to transport diverse cargo to the appropriate subcellular locations. Whereas axonal MTs are organized such that the plus-end is oriented out from the cell body, dendrites exhibit a mixed MTs polarity containing both minus-end-out and plus-end-out MTs. The molecular mechanisms underlying this differential organization, as well as its functional significance, are unknown. Here, we show that kinesin-1 is critical in establishing the characteristic minus-end-out MT organization of the dendrite in vivo. In *unc-116* (kinesin-1/kinesin heavy chain) mutants, the dendritic MTs adopt an axonal-like plus-end-out organization. Kinesin-1 protein is able to cross-link anti-paralleled MTs in vitro. We propose that kinesin-1 regulates the dendrite MT polarity through directly gliding the plus-end-out MTs out of the dendrite using both the motor domain and the C-terminal MT-binding domain.

**\*For correspondence:** kangshen@stanford.edu

**Competing interests:** The authors declare that no competing interests exist

**Reviewing editor**: Franck Polleux, Scripps Research Institute, United States

## Introduction

Neurons are highly polarized cells that usually elaborate two sets of morphologically and functionally distinct processes: a single long axon that sends out information to its targets, and multiple shorter dendrites that receive synaptic input from the environment or other neurons. Among the many differences between axons and dendrites, one distinct feature is that they have different microtubule (MT) organization. MTs, composed of strands of tubulin polymers, are dynamic structures with two molecularly and functionally distinct ends: a plus-end that favors polymerization and supports net growth and a minus-end that favors depolymerization (*Howard and Hyman, 2003*). Since MTs in neurites provide the platform for intracellular transport, and molecular motors such as kinesins and dyneins migrate toward either the plus- or minus-ends, MT polarity might play an important role in determining the directionality of transport events. One way to measure MT polarity in vivo is to perform time-lapse imaging studies using the EB family members of the TIPs (Tip Interacting Proteins), which bind transiently to the growing plus-ends of MTs (*Baas and Lin, 2011*). Based on the behavior of EB1 in neurites, it was cohesively concluded that MTs are uniformly polarized with their plus-ends oriented toward the distal tip in axons from worm to rodents. In the dendrites, MTs are mixed in the proximal region and uniformly plus-end-out in distal parts in cultured vertebrate neurons, whereas MTs exhibit a mixed orientation with minus-ends predominantly facing the distal dendrite in worm and fly (*Baas et al., 1987*; *Baas et al., 1988*; *Stepanova et al., 2003*; *Rolls et al., 2007*; *Stone et al., 2008*; *Kollins et al., 2009*; *Kapitein and Hoogenraad, 2011*; *Maniar et al., 2012*).

**eLife digest** Neurons, or nerve cells, are excitable cells that transmit information using electrical and chemical signals. Nerve cells are generally composed of a cell body, multiple dendrites, and a single axon. The dendrites are responsible for receiving inputs and for transferring these signals to the cell body, whereas the axon carries signals away from the cell body and relays them to other cells.

Like all cells, nerve cells have a cytoskeleton made up of microtubules, which help to determine cellular shape and which act as 'highways' for intracellular transport. Microtubules are long hollow fibers composed of alternating α- and β-tubulin proteins: each microtubule has a 'plus'-end, where the β subunits are exposed, and a 'minus'-end, where the α subunits are exposed. Nerve cells are highly polarized: within the axon, the microtubules are uniformly oriented with their plus-ends pointing outward, whereas in dendrites, there are many microtubules with their minus-ends pointing outward. This arrangement is conserved across the animal kingdom, but the mechanisms that establish it are largely unknown.

Yan et al. use the model organism *Caenorhabditis elegans* (the nematode worm) to conduct a detailed in vivo analysis of dendritic microtubule organization. They find that a motor protein called kinesin-1 is critical for generating the characteristic minus-end-out pattern in dendrites: when the gene that codes for this protein is knocked out, the dendrites in microtubules undergo a dramatic polarity shift and adopt the plus-end-out organization that is typical of axons. The mutant dendrites also show other axon-like features: for example, they lack many of the proteins that are usually found in dendrites. Based on these and other data, Yan et al. propose that kinesin-1 determines microtubule polarity in dendrites by moving plus-end-out microtubules out of dendrites.

These first attempts to explain, at the molecular level, how dendritic microtubule polarity is achieved in vivo could lead to new insights into the structure and function of the neuronal cytoskeleton.

Unlike in nonneuronal cells, where most MTs extend out from the microtubule organization center (MTOC) that harbors the minus-ends, the majority of neuronal MTs have 'free-floating' ends, raising the question of whether these microtubules are locally polymerized or transported from the cell body (*Baas and Lin, 2011*). Based on direct observation of short MT strand movement in cultured neurons, Baas and colleagues proposed a model in which the MT-based motors dynein and several mitotic kinesins directly slide MT strands into the axon and dendrite (*Baas, 1999*). More recently, Jan and colleagues showed that dynein was required for the uniform plus-end-out MT organization in *Drosophila* sensory neuron axons (*Zheng et al., 2008*). In dynein mutants, about 30% of the axonal MTs exhibit a minus-end-out orientation, whereas the dendritic MT polarity remains intact. Another recent study showed that the axon-localized MT-binding protein UNC-33/CRMP plays an important role in orienting dynamic MTs in both axons and dendrites in *Caenorhabditis elegans* neurons (*Maniar et al., 2012*). In *unc-33* mutants, both the axon and dendrite exhibit polarity defects. However, how predominant minus-end-out MT polarity is established in the dendrite remains largely unknown.

The signature pattern of MT polarity provides structural characteristics for axons and dendrites. In addition, the polarity pattern of MTs likely instructs the transport of motor-based polarized cargo and distribution of asymmetrical cellular material in the neuron. Thus, MT polarity might play an instrumental role in the establishment of neuronal polarity (*Baas, 2002*; *Hoogenraad and Bradke, 2009*). Here, we show that kinesin-1 is critical in generating the characteristic minus-end-out MT organization of the dendrite in vivo. Kinesin-1 (previously called kinesin heavy chain or KHC) is a plus-end-directed motor and has been reported to transport various cargoes including mitochondria, synaptic vesicles, and mRNAs (*Vale, 2003*; *Hirokawa and Takemura, 2005*). Like other kinesin family members, kinesin-1 contains an N-terminal motor domain, a coiled-coil region for dimerization, and a C-terminal tail domain for cargo binding and regulation (*Vale, 2003*; *Adio et al., 2006*). It has been showed that the kinesin-1 tail domain directly binds to MTs and mediates MT sliding in vitro (*Navone et al., 1992*; *Andrews et al., 1993*; *Jolly et al., 2010*; *Seeger and Rice, 2010*). In *unc-116* (kinesin-1/kinesin heavy chain) mutants, we find that dendritic MT polarity is completely reversed and adopts an axonal-like plus-end-out organization. The consequences of this polarity reversal in the dendrite include ectopic

accumulation of synaptic vesicles (SVs) and active zone proteins, and loss of dendritic enrichment of dendritic proteins. These results demonstrate that the proper polarized MT organization is essential for conferring neurite identity because it specifies the compartmental distribution of vital axonal and dendritic constituents such as SVs and neurotransmitter receptors in vivo. We also provide evidence that kinesin-1 is able to cross-link anti-parallel MTs in vitro. Combined with the structure–function analyses, we propose that kinesin-1 regulates the dendrite MT polarity through directly gliding the plus-end-out MTs out of the dendrite using both the motor domain and the C-terminal MT-binding domain.

## Results

### MT polarity in *C. elegans* neurons

In order to understand the molecular mechanisms and physiological importance of differential neuronal MT organization, we studied MT organization in two bipolar neurons in *C. elegans.* The ventrally localized DA9 cell body elaborates two functionally distinct processes. An axon migrates to the dorsal nerve cord, where it extends anteriorly and forms presynaptic specializations, and a dendrite extends anteriorly along the ventral nerve cord (*Figure 1A*; *White et al., 1976*; *Sym et al., 1999*; *Klassen and Shen, 2007*). MT motors fused to a fluorescence tag have previously been used as indicators for MT polarity (*Clark et al., 1997*; *Stone et al., 2008*). To examine MT polarity in the DA9 neuron, we expressed fluorescent proteins fused with retrograde and anterograde motors, which should accumulate at the minus- and plus-ends of MTs, respectively. We found that YFP-tagged UNC-104/Kinesin3 exhibits dim diffuse fluorescence along the axon with dramatic accumulation at the distal tip, but is completely absent from the dendrite (*Figure 1B*). Conversely, DHC-1/dynein heavy chain fused with GFP shows weak staining along the dendrite with accumulation at the distal tip of the dendrite (*Figure 1C*). In addition, GFP fused to the slow Ncd family minus-end kinesin, KLP-16, exhibits a graded fluorescence signal from proximal to distal dendrite with accumulation at the distal tip of the dendrite (*Figure 1D*). The specific localization of the tagged KLP-16 motor is dependent on its motor activity, as motor-dead versions are largely diffuse through the DA9 processes (*Figure 1—figure supplement 1*). These localization data are consistent with previous observations made in mammalian and *Drosophila* neurons, which suggest that MTs are uniformly plus-end-out in the axon and predominantly minus-end-out in the dendrite.

To extend these findings to other neurons, we used similar markers to examine MT polarity in the PHC sensory neurons. The two PHC neurons are bipolar with posteriorly oriented dendrites and anteriorly guided axons (*Figure 1K*). Consistent with the expression pattern in DA9, the plus-end kinesin UNC-104::YFP is enriched at the axonal tip of PHC, whereas the minus-end kinesin KLP-16::YFP decorates the dendrite and shows accumulation in the tip of dendrite (*Figure 1L* and data not shown).

The above data provide information regarding the overall steady-state polarity of MTs in neurites. To examine the polarity of dynamic MTs in the DA9 and PHC neurons, we fluorescently tagged the microtubule plus-end-tracking protein EBP-2/EB1. We observed characteristic 'comet' like movements of EBP-2::GFP in the dendrite and axon of both cell types by time-lapse analysis . The majority of the EBP-2::GFP comets in the axon move away from soma, whereas the majority of comets in the dendrite move toward the soma (*Figure 2A–D* and *Videos 1–3*). Consistent with published results (*Maniar et al., 2012*), these data show that axonal MTs are predominantly plus-end-out and dendritic MTs are largely minus-end-out in wild-type *C. elegans* neurons.

### UNC-116/kinesin-1 is required for minus-end-out MT organization in the dendrite

To understand how the minus-end-out MTs are assembled and maintained in the dendrite, we examined the localization of our MT markers in the mitotic kinesin mutant *zen-4/kinesin-6*, which is required for mobilizing MTs in neuronal processes in vitro (*Baas et al., 2006*). To our surprise, we found no obvious changes in the distribution of MT polarity markers in these mutants. Instead, we found that the DA9 dendritic MT organization is altered in mutants of a related kinesin, *unc-116*/kinesin-1. In *unc-116(e2310)* mutants, UNC-104::YFP also accumulates in the tip of dendrite in addition to its normal localization at the axonal tip (*Figure 1E*). Furthermore, the fluorescence signals of DHC-1::GFP and KLP-16::YFP no longer accumulate at the dendrite tip; instead, fluorescence is visible only at the dendritic segment nearest the cell body (*Figure 1F,G,M*). These changes in the distribution of MT polarity markers suggest that dendritic MT organization has changed to an axon-like plus-end-out

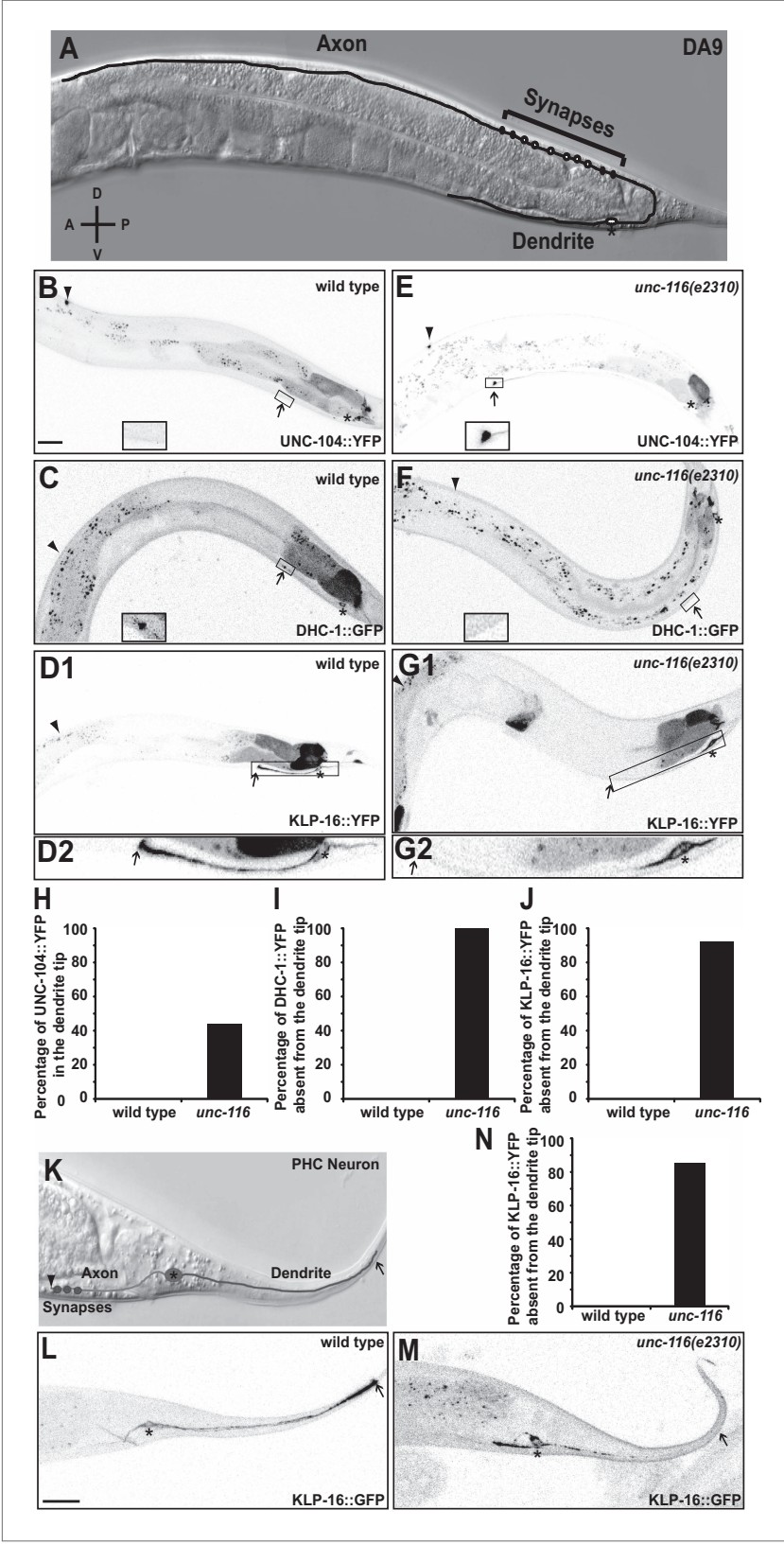

**Figure 1**. UNC-116 (kinesin-1) is required for the minus-end-out MT polarity in the DA9 dendrite. (**A**) Schematic diagram of the morphology of the DA9 neuron. Asterisk denotes DA9 or PHC cell body (throughout all images); D: dorsal; V: ventral; A: anterior; P: posterior. (**B**) and (**E**) Localization of UNC-104::YFP in a representative wild-type (**B**) or *unc-116*

*Figure 1. Continued on next page*

*Figure 1. Continued*

worm (**E**). Note that UNC-104::YFP is enriched in the axonal tip (denoted by an arrowhead throughout) in wild-type, while it is enriched at the tips of both the axon and dendrite (dendritic tip is marked by an arrow, dashed black box and shown in higher magnification micrographs throughout) in *unc-116* animals. (**C**) and (**F**) Localization of DHC-1::GFP in a representative wild-type (**C**) or *unc-116* worm (**F**). (**D**) and (**G**) Localization of KLP-16::YFP in a representative wild-type (**D1**) or *unc-116* worm (**G1**). The dendrite is shown in higher magnification for a wild-type (**D2**) or unc-116 animals (**G2**). (**H**)–(**J**) Quantification of fractions of worms with qualitative defects in UNC-104::YFP (**H**), DHC-1::GFP (**I**), and KLP-16::YFP(**J**) localization in DA9 dendrite (n > 50 for each genotype). (**K**) Schematic diagram of the morphology of the PHC neuron. (**L**) and (**M**) Localization of KLP-16::YFP in a representative wild-type (**L**) or *unc-116* worm (**M**). (**N**) Quantification of fractions of worms with qualitative defects in KLP-16::YFPlocalization in the PHC dendrite (n > 50 for each genotype). The scale bar represents 10 µm.

The following figure supplements are available for figure 1:

**Figure supplement 1**. Kinesin motor activity is required for the enrichment of KLP-16::YFP in dendrite.

**Figure supplement 2**. DA9 dendrite MT polarity is altered in *unc-116* mutants.

**Figure supplement 3**. UNC-116 acts cell autonomously in DA9 neuron.

pattern in the *unc-116* mutants. Consistently, in the same mutant dynamic MTs in the dendrites of both DA9 and PHC neurons almost completely change their behaviors, adopting a predominantly plus-end-out movement (*Figure 2A–C* and *Videos 4 and 5*). These data further indicate that the *unc-116*/kinesin-1 mutation causes MT polarity in dendrites to become axon-like, but with no effect on MT orientation in the axon.

UNC-116 encodes the sole *C. elegans* ortholog of kinesin-1 (previously called kinesin heavy chain or KHC) (*Siddiqui, 2002*). Since complete loss of *unc-116* leads to embryonic lethality and *unc-116(e2310)* is viable, the *e2310* allele likely represents a partial loss-of-function mutant. Two additional partial loss-of-function alleles of *unc-116, wy270* and *rh24sb79,* show similar mislocalization of MT motors in the DA9 dendrite, suggesting that this phenotype is indeed due to loss of the protein activity (*Figure 1—figure supplement 2*). Consistent with this notion, cell autonomous expression of wild-type UNC-116 in *unc-116(e2310)* mutants rescued the dendritic distribution phenotype of KLP-16, indicating that UNC-116 functions in the DA9 neuron (*Figure 1—figure supplement 3*). Taken together, these data support the idea that UNC-116's function is required for establishing the minus-end-out MT organization in the dendrite.

## Synaptic vesicle and active zone proteins localize to dendrites in *unc-116* mutants

What are the consequences of MT polarity reversal in the DA9 dendrite? We first compared the development of the DA9 dendrite in wild-type and *unc-116* animals. In wild-type animals, DA9 axonal outgrowth and neuromuscular junction formation occur embryonically, whereas most dendrite outgrowth takes place postembryonically starting from the L1 stage (*Teichmann and Shen, 2011*). Upon reaching adulthood, the DA9 axon and dendrite achieve stereotyped lengths. In the *unc-116(e2310)* mutants, the sequence and directionality of axonal and dendrite outgrowth as well as the length of the axon are indistinguishable from that of the wild-type strain. The length of the DA9 dendrite, however, is dramatically longer compared with wild-type animals at every stage of development (*Figure 3—figure supplement 1*). To determine whether MT-dependent vesicle transport is affected by *unc-116(e2310)*, we examined localization of synaptic vesicle (SV) markers. The UNC-104/Imac/kinesin3 motor traffics SVs to presynaptic specializations (*Hall and Hedgecock, 1991*), and synaptic vesicle proteins like RAB-3 exclusively accumulate in the axon of wild-type animals (*Figure 3A*). In *unc-104(e1265)* mutants, SV markers are completely absent from the axon, accumulating instead in the cell body and dendrite due to the activity of dynein (*Figure 3C*; *Ou et al., 2010*). Interestingly, we found that in *unc-116(e2310)* mutants, both the axon and dendrite contain a similar number of vesicle clusters (*Figure 3E* and *Figure 3—figure supplement 2*). If this ectopic accumulation of SVs in the dendrite is due to the switch of MT polarity to axon-like plus-end-out arrangement (rather than to the activity of dynein), then the accumulation of SVs in the DA9 dendrite should depend on the UNC-104

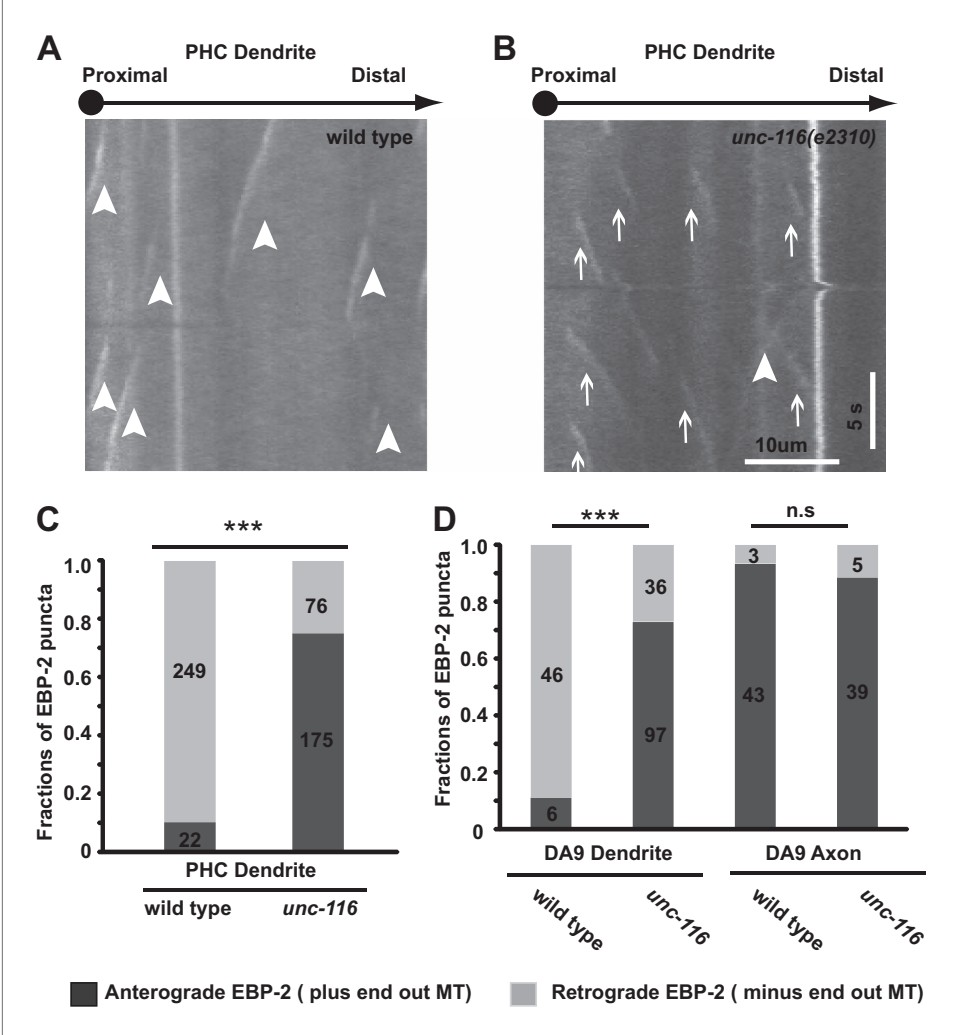

**Figure 2**. Dendrite MTs are plus-end out in the *unc-116(e2310)* mutant. (**A**) and (**B**) Representative kymographs of moving EBP-2::GFP puncta in the PHC neuron dendrite of wild-type (**A**) and *unc-116* animals (**B**). The cell body is to the left in both panels. Time runs top to bottom; arrowheads mark retrogradely moving puncta; arrows mark anterogradely moving puncta. (**C**) and (**D**) Bar graphs of the fraction of anterograde and retrograde movements in PHC (**C**) and DA9 neurons (**D**). MT, microtubule; numbers within each column denote the number of puncta counted in the corresponding categories; ***p<0.001, $\chi^2$ test.

kinesin motor. Indeed, in *unc-104(e1265);unc-116(e2310)* double mutants, SVs are largely absent from both processes and trapped in the cell body. These results argue that UNC-116 is not directly involved in SV transport in the DA9 neuron; rather, it creates the minus-end-out MT polarity of the dendrite. As a result of this aberrant MT polarity, the plus-end motor UNC-104 promotes locomotion of cargoes into the dendrite. If our model is correct, the *unc-116* mutations should not only affect RAB-3 but also other synaptic vesicle proteins and active zone proteins. Indeed, the active zone protein SYD-2/liprin and other SV markers like SNG-1/synaptogyrin show similar ectopic localization to DA9 dendrites in the *unc-116(e2310)* mutants, suggesting that the changes in MT polarity affect the entire presynaptic structure (***Figure 3—figure supplement 2***).

## The dendrite fails to accumulate dendritic proteins in *unc-116* mutants

To determine whether *unc-116* and dendritic MT polarity are critical for localization of dendritic proteins, we analyzed the distribution of several dendritically targeted proteins. An acetylcholine receptor (ACR-2), a receptor tyrosine kinase (CAM-1), and a gap junction protein (FBN-1) are all reported to be dendritically enriched in DA9 (***Sieburth et al., 2005***; ***Barbagallo et al., 2010***). In wild-type

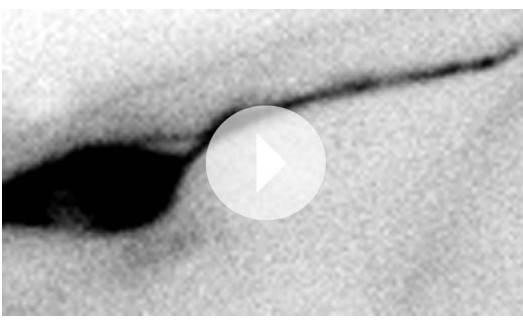

**Video 1**. Movement of EBP-2::GFP puncta in the ventral axon of wild-type DA9 neuron. Cell body is to the left. Displayed 10× speed.

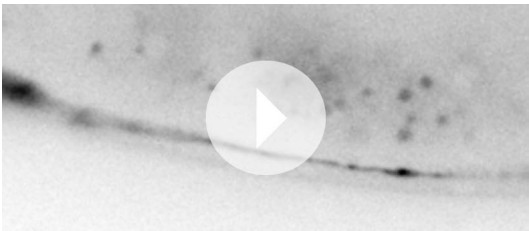

**Video 2**. Movement of EBP-2::GFP puncta in the dendite of wild-type DA9 neuron. Cell body is to the left. Displayed 10× speed.

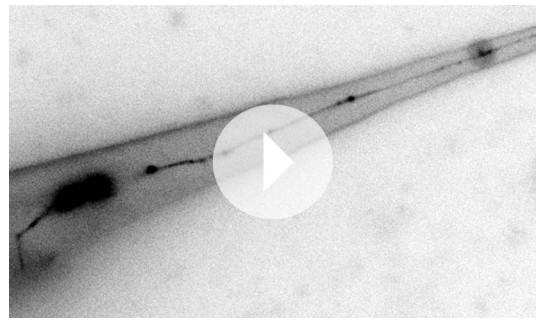

**Video 3**. Movement of EBP-2::GFP puncta in the dendrite of wild-type PHC neuron. Cell body is to the left. Displayed 10× speed.

DA9 neurons, fluorescently tagged ACR-2, CAM-1, and FBN-1 all localize to the dendrite, cell body, and the initial part of the ventral axon. In *unc-116(e2310)* animals, however, these dendritic proteins accumulate within the cell body and largely disappear from the dendrite, suggesting that *unc-116* is required for dendritic localization of postsynaptic proteins (*Figure 4*). Taken together, these data suggest that the minus-end-out MT organization in the dendrite is instructive for dendritic cargo transport by molecular motors. It discourages the plus-end motor, UNC-104, and encourage the minus-end motor, dynein, from entering and accumulating in the dendrite. This polarity mechanism is thus essential for setting up the polarized distribution of vital axonal and dendritic constituents such as SVs and neurotransmitter receptors.

## Kinesin-1 cross-links anti-parallel MTs in vitro

The canonical function of kinesin-1 is to transport certain cargos, such as mitochondria, along neuronal MTs. Like kinesin-1, UNC-116 is composed of an N-terminal motor domain, a putative coiled-coil dimerization stalk region, and a C-terminal tail domain that binds to kinesin light chains (KLCs), adaptors, and cargos (*Vale, 2003*). Interestingly, several recent biochemical and cell biology experiments have shown that part of the C-terminal tail domain of kinesin-1 binds directly to MTs (*Navone et al., 1992*; *Dietrich et al., 2008*; *Seeger and Rice, 2010*). It was also demonstrated that kinesin-1 could cross-link and glide MTs using in vitro assays (*Andrews et al., 1993*; *Palacios and St Johnston, 2002*; *Yamada et al., 2010*). To understand how UNC-116 regulates dendrite MT polarity, we considered two possible models in which UNC-116/kinesin-1 directly transport the minus-end-out MTs into dendrites or transport the plus-end-out MTs out of dendrites via interaction of its C-terminal MT-binding domain with the cargo MTs. In either scenario, it requires kinesin-1 selectively cross-link anti-parallel MTs. We first tested it using in vitro MTs bundling assay. When purified tubulins were polymerized into MTs, the MTs were found as isolated microtubules with the appearance of smooth surface under the electron microscope (*Figure 5A*). Addition of purified kinesin-1 complex from bovine brain to the MTs causes the dispersed MTs to adopt a more 'rough' surface due to the binding of kinesin-1 to MTs (*Figure 5B*). When both of kinesin-1 and mNUDC were added to the MTs, we found that MTs were bundled (*Figure 5C*). The MT bundles are superficially similar to those reconstructed from flagellar axonemal dyneins and MTs (*Ueno et al., 2008*). No such bundles were observed in control samples containing just taxol-stabilized MTs or in mixtures of MTs with kinesin-1 only (*Figure 5B*). Neighboring MTs in the bundles has an interval of ~80 nm. Considering kinesin-1 has a long flexible stalk (maximum ~80 nm; *Hirokawa et al., 1989*) and mNUDC is an adapter protein, it is conceivable that the cross-bridges represent a complex of kinesin-1 and mNUDC. Furthermore, we determined the polarity of the cross-linked MTs from their

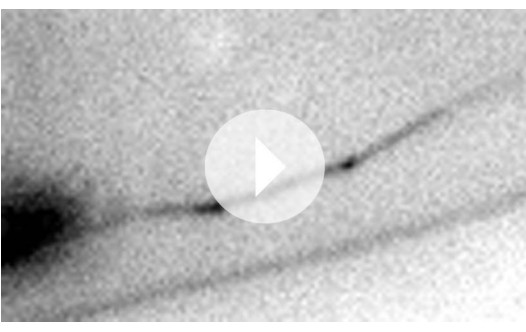

**Video 4**. Movement of EBP-2::GFP puncta in the ventral axon of unc-116 (e2310) DA9 neuron. Cell body is to the left. Displayed 10× speed.

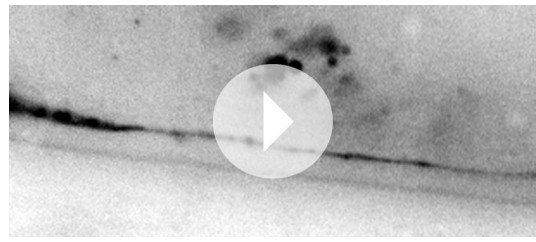

**Video 5**. Movement of EBP-2::GFP puncta in the dendrite of unc-116 (e2310) DA9 neuron. Cell body is to the left. Displayed 10× speed.

Moiré patterns of protofilaments (*Chretien et al., 1996*), and determined orientation of 13 pairs of MTs cross-linked by kinesin-1. Nine pairs (69%) of microtubule bundles were anti-parallel with a 95% confidence interval (CI) of 39% and 91% by *F*-test for binominal distribution (*Figure 5D–D2*). The lower limit of CI exceeds 50% when a significant level is 13.4% using one-tailed *F*-tests. Therefore, kinesin-1-mNUDC complex tends to cross-link anti-parallel microtubule bundles.

## UNC-116/kinesin-1 slides plus-end-out MTs out of the dendrite through its C-terminal MT-binding site

The ability of UNC-116 to slide anti-parallel MTs requires that the motor have two distinct binding sites for MTs. Indeed, the C-terminal domain of kinesin-1 has been shown to bind to MTs in addition to the N-terminal motor domain (*Navone et al., 1992*; *Seeger and Rice, 2010*). To further investigate whether the C-terminal domain of UNC-116 can bind to MTs and whether this binding is required for its function in establishing dendritic MTs, we first examined the amino acid sequence of the C-terminal region of UNC-116. Sequence alignment of UNC-116 with its corresponding fly, rat, and human homologs showed that the MT-binding site is highly conserved (*Figure 6A*). The positively charged residues (labeled in red) in this region have been shown to be required for MT-binding in vitro (*Seeger and Rice, 2010*). Given the high degree of sequence conservation of the MT-binding site, it is likely that its function in binding MTs is also conserved in worms. Furthermore, GFP-tagged C-terminal domain of UNC-116 (521–863) showed colocalization with MTs when expressed in HEK293 cells, suggesting that the C-terminal domain of UNC-116 can indeed bind to MTs (data not shown).

Second, we examined the molecular lesions in three *unc-116* mutants, all of which show MT polarity defects (*Figure 1—figure supplement 3*). The molecular lesion of *e2310* is a TC5 transposon insertion. The transposon was inserted after residue 692, which truncates the C-terminal MT-binding domain (*Patel et al., 1993*). Point mutations are found in the motor domain in wy270 and rh24sb79 alleles, which likely result in partial loss of motor function (*Yang et al., 2005* and unpublished data). These results suggest that both motor activity and C-terminal MT-binding might be required for this newly defined function of UNC-116/kinesin-1. To definitively test this, we created a mutant form of UNC-116 in which three point mutations were introduced in the C-terminal MT-binding motif. The same mutations diminish the binding of kinesin-1 C-terminal domain to MTs (*Seeger and Rice, 2010*). Therefore, this mutant form (UNC-116$_{MT}$) might specifically lack its ability to bind to MTs with the C-terminal motif. We tested whether this form can still rescue the dendrite polarity phenotypes as well as phenotypes related to the canonical transport function of UNC-116. To assess the canonical organelle transport function of UNC-116, we examined localization of mitochondria in DA9. In wild-type animals, a mitochondrial marker TOM-20::YFP localizes to discrete puncta in both the axon and dendrite of DA9 (*Figure 6—figure supplement 1*). In *unc-116(e2310)* mutants, the TOM-20 signal is completely absent from both processes and only found in the cell body (*Figure 6—figure supplement 1*), suggesting that UNC-116 is required for transporting mitochondria to both axon and dendrite. DA9 expression of UNC-116$_{MT}$ efficiently rescues the TOM-20 localization defect in the axon, suggesting that this mutant form of UNC-116 can support the organelle transporting function. Interestingly, but the same transgene showed much less rescue of the MT polarity defects in the dendrite, as assayed by KLP-16 localization (*Figure 6B*). This result is consistent with the notion that the C-terminal MT-binding motif

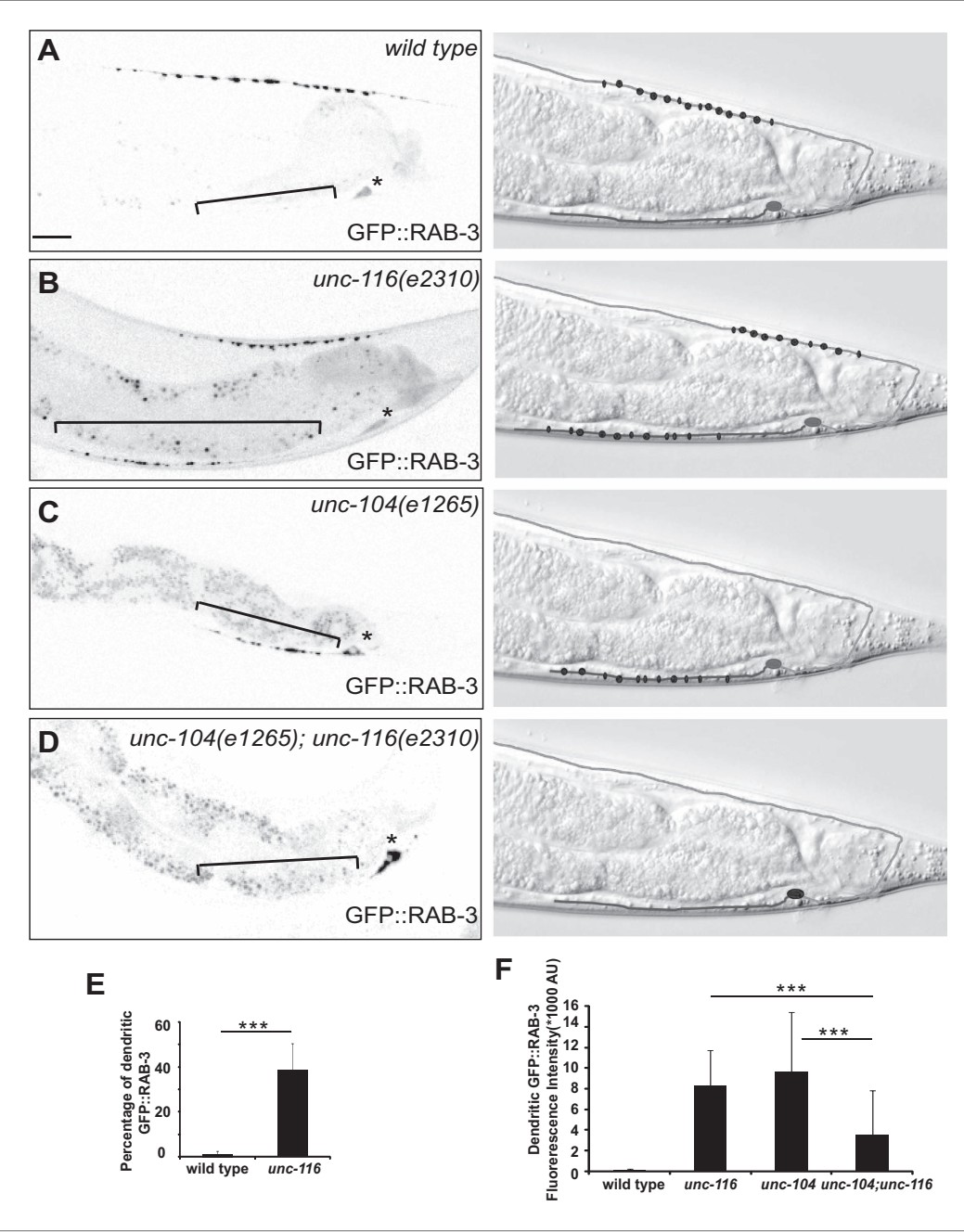

**Figure 3**. Dendrite exhibits axonal-like properties in *unc-116* mutants. (**A**)–(**D**) Distribution of GFP::RAB-3 puncta (left panels) and represented schematic diagrams (right panels) in wild-type (**A**), *unc-116* (**B**), *unc-104* (**C**), and *unc-116;unc-104* mutant animals (**D**). (**E**) Average percentage of GFP::RAB-3 fluorescence intensity in the dendrite (n = 20). (**F**) Quantification of GFP::RAB-1 fluorescence intensity in the dendrite (n = 20). ***p<0.0001. Student's *t*-test. The scale bar represents 10 μm. Asterisk denotes DA9 cell body.

The following figure supplements are available for figure 3:

**Figure supplement 1**. DA9 dendrite is longer in *unc-116* mutants.

**Figure supplement 2**. Presynaptic proteins are mis-localized in *unc-116* mutant.

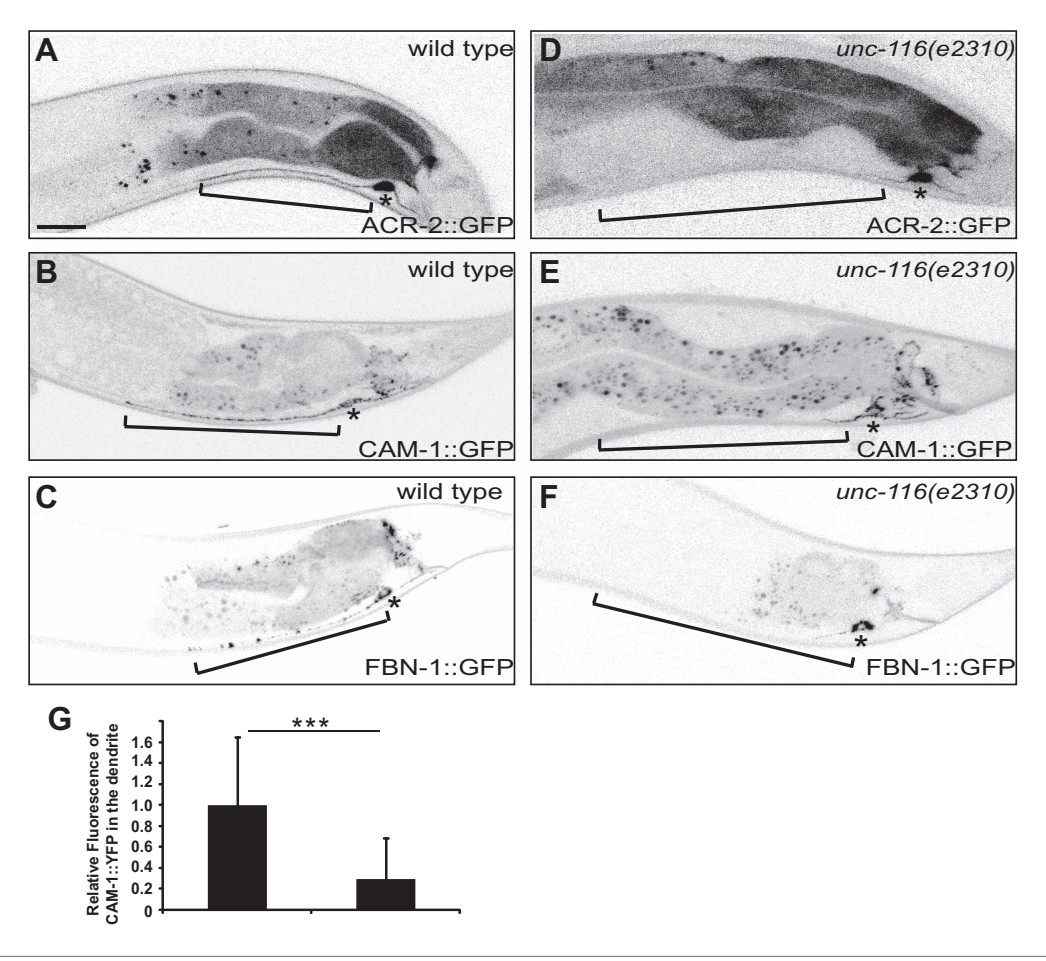

**Figure 4**. DA9 dendrite fails to accumulate dendritic proteins in *unc-116* mutants. (**A**) and (**D**) Distribution of the postsynaptic neurotransmitter receptor ACR-2::GFP in wild-type (**A**) and *unc-116* (**D**) animals. Bracket indicates dendrite region. (**B**) and (**E**) Localization of CAM-1::YFP in a representative wild-type (**B**) or *unc-116* worm (**E**). (**C**) and (**F**) Localization of FBN-1::YFP in a representative wild-type (**C**) or *unc-116* worm (**F**). (**G**) Quantification of CAM-1::YFP fluorescence intensity in the dendrite (n = 20). \*\*\*p<0.0001. Student's *t*-test. The scale bar represents 10 μm. Bracket indicates dendrite. Asterisk denotes DA9 cell body.

of UNC-116 is specifically required for the establishing dendritic MT polarity. In addition, UNC-116$_{MT}$ efficiently rescues the abnormally long dendrites in the *unc-116(e2310)* mutant, suggesting that this length phenotype is independent of the MT polarity.

To further explore the difference between the MT-polarity-related function of UNC-116 and its canonical vesicular and organelle transport function, we studied the kinesin light chains, KLC-1 and KLC-2. Consistent with previous reports, KLC-2 is required for axonal localization of TOM-20, suggesting that the KLCs are required for the organelle transport function of UNC-116 (*Figure 6— figure supplement 1*). Interestingly, neither *klc-1* or *klc-2* single mutants nor *klc-1*; *klc-2* double mutants showed any defects in MT polarity markers (*Figure 6—figure supplement 1*), suggesting that they are dispensable for the MT polarity function of UNC-116. Indeed, the MT-gliding activity of kinesin-1 in nonpolarized *Drosophila* S2 cells is also independent of KLCs, suggestive of a similar mode of action in nonneuronal cells (*Jolly et al., 2010*). Taken together, our structure–function analyses have revealed a novel function of UNC-116 in regulating dendritic MT polarity. This function requires both motor activity and C-terminal MT binding, and can be separated from the canonical vesicular and organelle trafficking function of UNC-116 both at the level of cargo-binding domains as well as the dependency on kinesin light chains.

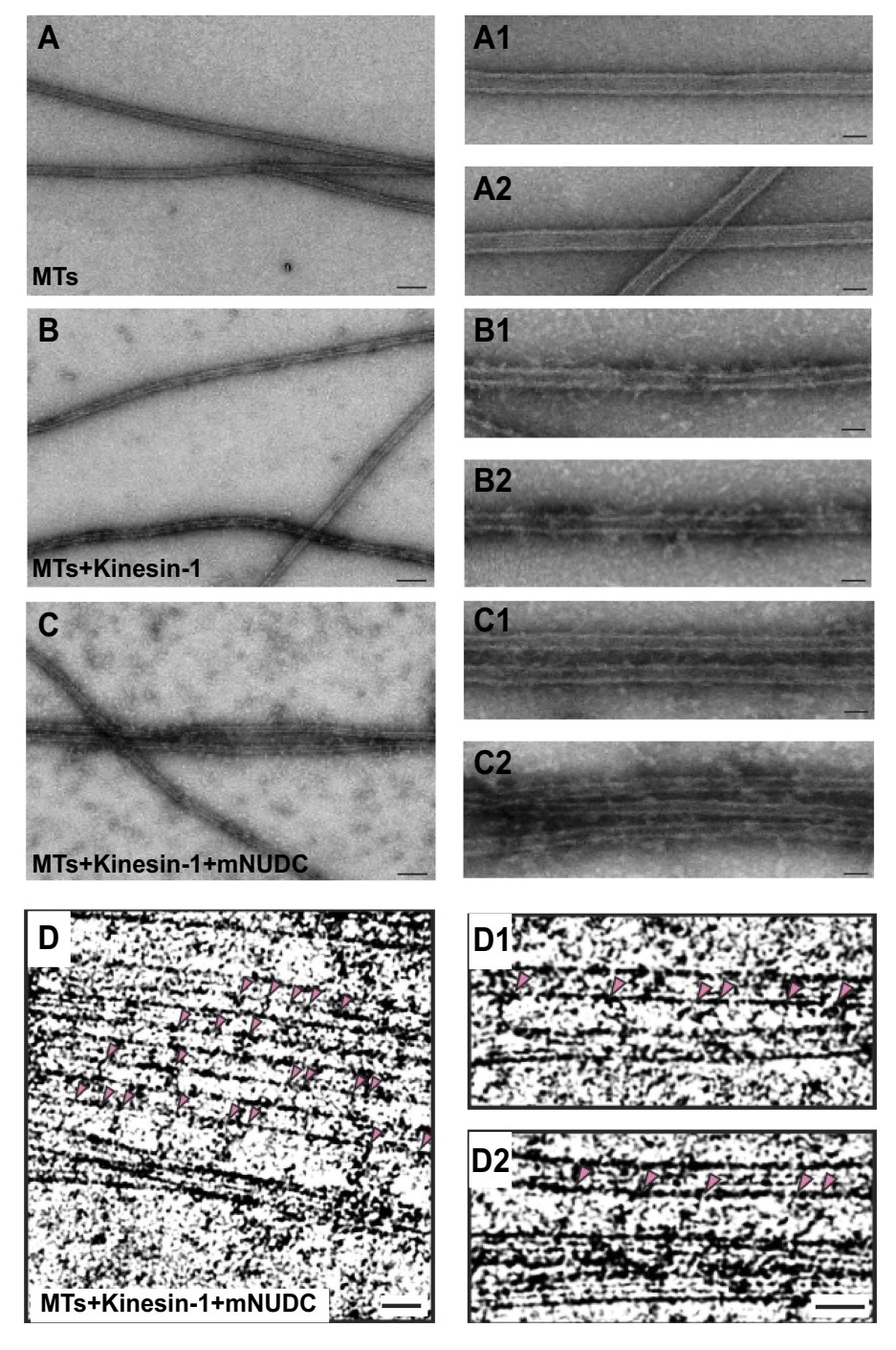

**Figure 5**. Purified kinesin-1 complex bundles anti-parallel MTs in vitro. (**A**)–(**C**) Negatively stained EM images of (**A**) MTs alone, (**B**) kinesin-1-MTs, and (**C**) kinesin-1-NudC-MTs. Two enlarged images are in right row. Scar bar is 100 nm from (**A**–**C**), and 30 nm in the enlarged images. Note that long spanned MT bundles were observed only in (**C**) when the samples mixed with both Kinesin-1 and NudC. Deeply stained cross-bridging molecules are seen between adjacent MTs. Cryo-EM images of microtubules of (**D**) kinesin-NudC microtubules, (**D1**) microtubules alone and (**D2**) kinesin-microtubules.

How does UNC-116/kinesin-1 promote minus-end-out MTs configuration in the dendrite? Our genetic data so far are consistent with two models. In the first model, UNC-116/kinesin-1 directly slides minus-end-out MTs into dendrites. Alternatively, UNC-116 might slide plus-end-out MTs out of dendrite and consequently increase the percentage of the minus-end-out MTs in dendrites. In

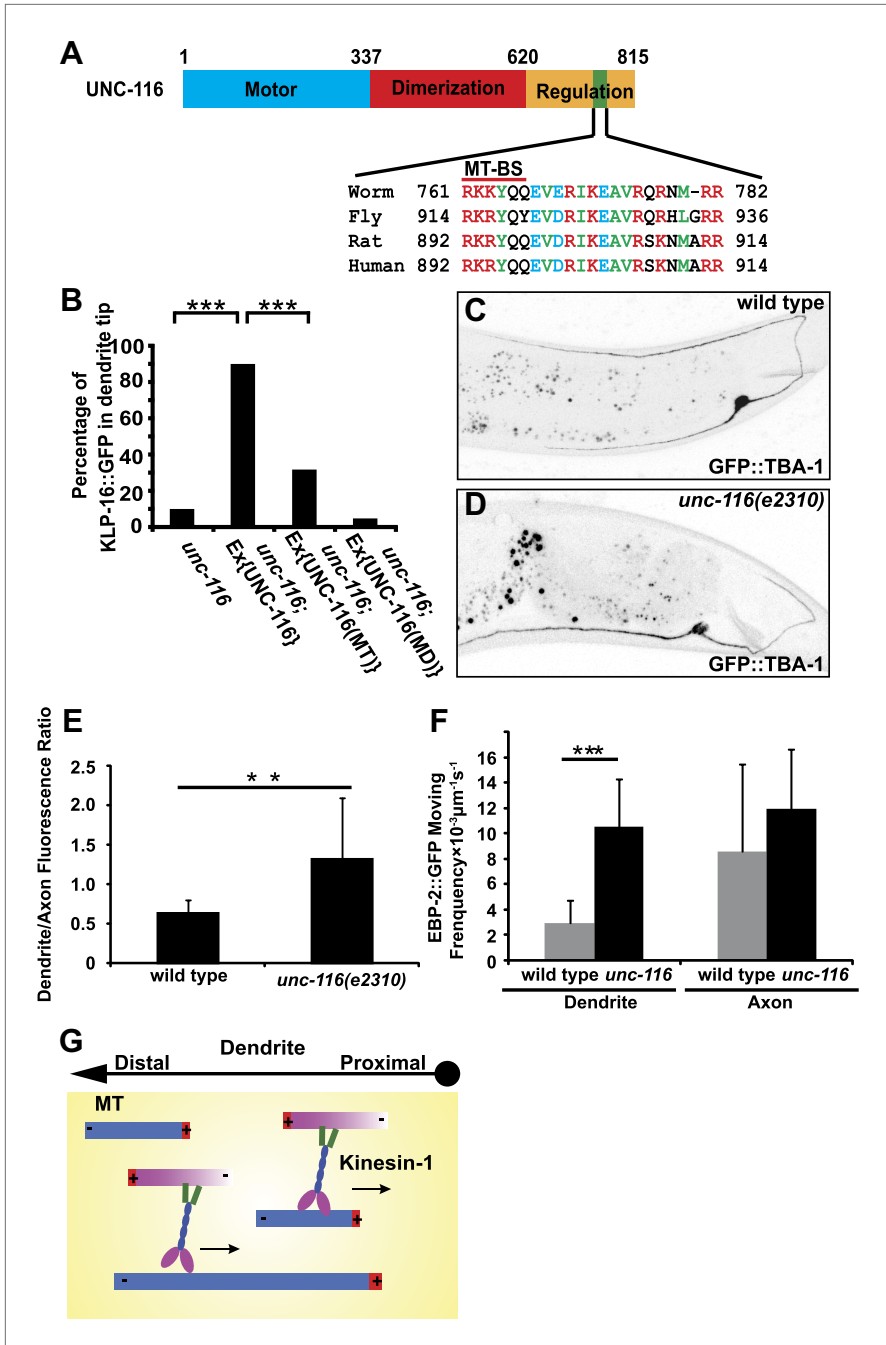

**Figure 6**. UNC-116/kinesin-1 orients dendritic MTs. (**A**) Domain structure of UNC-116 protein with motor domain in blue, dimerization region in red, and C-terminal regulation domain in orange, length of amino acid is shown. (**B**) Cell-autonomous rescue of KLP-16::YFP localization. MD, mutant swapping the amino acids 'KTH' in P-loop of UNC-116 motor domain with 'AAA'; MT, mutant swapping the MT-BS in UNC-116 tail region with 'AAAYAA'. n > 50 per group. Error bar indicates standard deviation; ***p<0.0001, $\chi^2$ test. (**C**) and (**D**) MTs structure highlighted by GFP::TBA-1 in a representative wild-type (**C**) or *unc-116* worm (**D**). Quantification of GFP::TBA-1 fluorescence density ratio between the dendrite and the ventral axon in wild-type and the *unc-116(e2310)* animals. (**E**) Quantification of EBP-2::GFP puncta moving frequency in the dendrite of wild type and the *unc-116(e2310)* animals. n > 20 per group. Error bar indicates standard deviation; **p<0.001, ***p<0.0001, Student's *t*-test. (**F**) A schematic model showing kinesin-1 cross-linking and sliding MTs out of the dendrite.

The following figure supplements are available for figure 6:

**Figure supplement 1**. Kinesin light chains are not required for dendrite MT polarity.

order to differentiate these two models, we analyzed the MTs concentration in the dendrite and axon. We reasoned that if UNC-116/kinesin-1 slides minus-end-out MTs into dendrite, there would be less MTs in the dendrite of mutant animals. If UNC-116/kinesin-1 slides plus-end-out MTs out of dendrite, there would be more MTs in the dendrite of mutant. We visualized the MTs structure by expressing GFP::TBA-1/ α-tubulin in the DA9 neuron. In wild-type animals, axon has higher expression of GFP::TBA-1, with a dendrite/ventral axon intensity ratio around 0.6. In *unc-116(e2310)* animals, axon expression of GFP::TBA-1 remains largely unchanged, whereas expression in the dendrite increases, with a dendrite/axon intensity ratio around 1.2, suggestive of excessive MTs in the *unc-116* mutant dendrites (*Figure 6C–E*). Another measure of the number of MTs is the frequency of EBP-2::GFP comets, which directly reflects the number of dynamic plus-ends. We found that EBP-2::GFP comets in the *unc-116* mutant DA9 dendrite is three times more frequent comparing to wild type, further indicating there are more MTs in *unc-116(e2310)* dendrite (*Figures 2C, 6F*). Collectively, these data favor the model that UNC-116/kinesin-1 maintains minus-end-out MT polarity in dendrite through selectively sliding plus-end-out MTs out of dendrite (*Figure 6G*).

## Discussion

We report a new function of kinesin-1 to promote the minus-end-out MT organization in the dendrite. Previous studies in neuronal cell cultures showed that intracellular transport of short MTs occurs robustly in neurites and can be driven by mitotic kinesins (*Baas et al., 2006*). While we did not find evidence that mitotic kinesins are required for DA9 MT organization, our data provide in vivo evidence that transport of MTs by another kinesin, kinesin-1, in the dendrite is critical for establishing or maintaining MT polarity. An interesting study in *Drosophila* sensory neurons showed that the minus-end motor dynein is required for the uniform plus-end-out MT organization in the axon but dispensable for dendritic MT organization (*Zheng et al., 2008*). A recent study in *Drosophila* neuron showed that another motor kinesin-2 complex is required to steer MTs growth in the dendrite branch point to maintain the uniform minus-end-out MT polarity in the dendrite (*Mattie et al., 2010*). Since kinesin-1 is highly conserved throughout evolution, we anticipate our findings to be a starting point for more sophisticated in vitro or in vivo analyses of MT polarity in neurons. If UNC-116/kinesin-1 slides plus-end-out MTs out of dendrites, one immediate question is how this process is restricted to dendrites. *Maniar et al. (2012)* recently showed that the MT-associated axon initial segment (AIS)-enriched protein UNC-33/CRMP is important for MT organization in both the axon and dendrite in worm sensory neurons. The AIS has been reported to play important roles in neuronal polarity (*Hedstrom et al., 2008*; *Sobotzik et al., 2009*). It is conceivable that AIS proteins provide critical regulatory roles for motor-based MT sliding by restricting the direction of MT transport. Interestingly, the AIS in vertebrate neurons is known to serve as a diffusion barrier for both transmembrane and cytosolic proteins. How it regulates MT transport will be an interesting question for future studies.

The compartmentalization of neurons is achieved by several cell biological mechanisms including directed intracellular trafficking, local sequestration, diffusion barriers, and transcytosis (*Kennedy and Ehlers, 2006*). Previous studies have also reported the existence of 'smart motors' that are capable of distinguishing axonal and dendritic MTs (*Saito et al., 1997*; *Marszalek et al., 1999*; *Guillaud et al., 2003*; *Kapitein et al., 2010*). The existing literature raises the question of what functional role is played by MT polarity, an important question that cannot be answered without a polarity-altering tool. The *unc-116* mutants specifically affect MT polarity in dendrites, and therefore serve as useful reagents to dissect the importance of dendritic MT polarity. Our data strongly support the notion that the minus-end-out MTs in dendrites not only allow for the accumulation of dendritic proteins but are also critical for the exclusion of axonal components such as synaptic vesicles and active zone proteins. Furthermore, MT polarity appears to affect the length of the dendrite, suggesting that both the structural and transport functions of MTs depend on their polarity. Interestingly, the direction and timing of dendrite outgrowth are not affected in *unc-116* mutants, suggesting that MT polarity does not play a critical role in dendrite outgrowth, a process in which the actin cytoskeleton is known to play an important role (*Gao et al., 1999*). Together, these data argue that polarized MT tracks play instructive roles in formation of axonal and dendritic identities, as well as in the compartmentalization of axonal and dendritic constituents such as SVs, active zone proteins, and neurotransmitter receptors.

## Materials and methods

### Strains and genetics

Worms were raised on NGM plates at 20°C using OP50 *Escherichia coli* as a food source. N2 Bristol was utilized as the wild-type reference strain. The following mutant strains were obtained through the *Caenorhabditics* Genetics Center: FF41 *unc-116(e2310) III*, RB1975 *klc-1(OK2609) IV*, and CB1265 *unc-104(e1265) II*. TV6687 *unc-116(rh24 sb79) III* was a kind gift from Dr. McNally at University of California at Davis, TV3706 *unc-116 (wy270) III*, carrying a single L129 to F mutation in motor domain, (EL, unpublished data) was previously isolated from our laboratory.

### Molecular biology and transgenic lines

Expression clones were made in the pSM vector, a derivative of pPD49.26 (A Fire, unpublished data) with extra cloning sites (S McCarroll and CI Bargmann, unpublished data). The plasmids and transgenic strains (1–50 ng/μl) were generated using standard techniques and coinjected with markers *Podr1::RFP* or *GFP* (40 ng/μl): wyEx3892 [*Pitr1::dhc-1::GFP*], wyEx 2559 [*Pitr1:: unc-104::YFP*], wyEx3128 [*Pitr1::klp-16::YFP*], wyIs674 (*Pitr1::klp-16::YFP*), wyIs309 (*Pmig13::klp-16::GFP*), wyEx4974 [*Pida-1::unc-104::GFP*], wyEx4980 [*Pida-1::klp-16::GFP*], wyEx4978 [*Pida-1::GFP::rab-3*], wyEx4828 [*Pdes-2::ebp-2::GFP*], wyIs349 (*Pmig13::loxP-ebp-2::GFP*), wyIs85 (*Pitr1::GFP::rab-3*), wyEx2505 [*Pitr1::syd-2::GFP*, *Pitr1::mCherry::rab-3*], wyIs386 (*Pitr1::acr-2::GFP*), wyEx1902 [*Pitr1::mCherry*], wyEx403 [*Pitr1::cam-1::YFP*], wyEx2396 [*Pitr1::fbn-1;;YFP*], and wyEx2709 [*Pitr1::tom-20::YFP*]. Detailed subcloning information will be provided on request.

### Fluorescence microscopy and confocal imaging

Images of fluorescently tagged fusion proteins were captured in live *C. elegans* using a Plan-Apochromat 63×/1.4 objective on a Zeiss LSM510 confocal microscope system. Visual inspections and some quantification were done using a Zeiss Axioplan 2 microscope with Chroma HQ filter sets for GFP, YFP, and RFP (63×/1.4NA objective). Worms were immobilized using 10 mM levamisole (Sigma, St Louis, MO) and oriented anterior to the left and dorsal up.

### Dynamic imaging

Dynamic imaging was performed on an inverted Zeiss Axio Observer Z1 microscope equipped with the highly sensitive QuantEM:512SC camera. A Plan-Apochromat 63×/1.4 objective was used for acquisition. L4 worms were cultured at 22°C for imaging. They were mounted onto 2% agarose pads and anesthetized with 6 mM levamisol (Sigma) for no longer than 20min. All videos were acquired over 25 s with 8 frames per second. Videos were then analyzed using the ImageJ software.

### Protein purification

Tubulin was purified from porcine brain and polymerized to microtubules by a previously described method, then stabilized by taxol (*Sloboda and Rosenbaum, 1982*). Conventional kinesin (kinesin-1) was also purified from porcine brain as described before (*Wagner et al., 1991*). Recombinant protein for mNUDC was generated using pGEX-4T expression vector (GE Healthcare Bio-sciences). Protein purification was performed using Glutathione Sepharose 4B (GE Healthcare Bio-sciences) based on the manufacturer's recommendation. To remove GST-tag, we treated recombinant protein with Thrombin (Merck), followed by absorption of thrombin by Benzamidine Sepharose 6B (GE Healthcare Bio-sciences).

### Negatively stained electron microscopy

Taxol-stabilized MTs (25 μg/ml) with/without kinesin-1 (85 μg/ml), NudC (50 μg/ml), and 2 mM AMP-PNP were mixed in a test tube according to the condition then sit for 10 min. The kinesin-1 and NudC were mixed at a molar ratio of 1:5. The mixed solution of 5 μl was loaded on a hydrophilized carbon grid (75/300 mesh; VECO, Cu) to sit for 60 s in a humid chamber. Unabsorbed protein was rinsed by BRB80 buffer (80 mM Pipes-KOH, 2 mM $MgSO_4$, 1 mM EGTA, pH 6.8) containing with 10 μM taxol. After the excess solution was blotted with a filter paper, the specimen was stained with an equal volume of 2% uranyl acetate for 40 s. The specimens were observed with a transmission electron microscope, Tecnai Spirit (FEI Co.). The microscope was operated at 120 kV. The magnification for overviewed images was ×21,000 and the nominal magnification was ×67,000. The defocusing values were usually −1 to −1.5 μm. The images were recorded using CCD (2k × 2k, Eagle1k CCD; FEI) at 30 μm/pixel corresponding to 0.47 nm. The electron dose per single image was estimated as 10 e/$Å^2$.

## Cryo-electron microscopy

To define the polarity of the MTs, electron cryo-microscopy was used. First frozen-hydrated specimens were prepared as follows: a 10-µl drop of purified MT/kinesin-1/NudC complex with 2 mM AMP-PNP was mounted on a holey carbon grid (Quantifoil: Mo R2/2, Germany) and blotted with filter paper to remove excess solution and make a thin aqueous layer. The grid was then plunged into ethane slush at −185°C to create a thin layer of vitreous ice. The frozen-hydrated specimens were examined at liquid nitrogen temperature using a cryo-holder (CT3500; Oxford Instruments, United Kingdom) with an EF-2000 microscope (Hitachi High-Technologies, Japan) operated at 200 kV. The electron micrographs were recorded using a 2 k × 2 k CCD camera developed by TVIPS (*Yasunaga and Wakabayashi, 2008*) with an defocusing value of −3 to −4 µm. The electron dose per single image was estimated as approximately ~1400 e/Å$^2$.

## Image analysis

The defocusing values of all obtained images were estimated by 'ctfDisplay'. The images were then corrected by phase flipping after the phase contrast transfer functions (CTF) were determined from the defocusing values. Corrected images were processed by the band-pass filter (1/100 nm to 1/10 nm), and so their background noise was reduced. The polarity of microtubules was determined by checking their Moiré patterns. Most of the image processing was performed by Eos (*Yasunaga and Wakabayashi, 1996*).

## Acknowledgements

This work was supported by the Howard Hughes Medical Institute. The authors thank the International *C. elegans* Gene Knockout Consortium, the National Bioresourse Project-Japan for strains. The authors also thank C Gao and B Tara for technical assistance, P Kurshan, C Richardson, PH Chia, E Stewart, and members of the Shen laboratory for thoughtful comments on the manuscript. J Yan is supported by the Human Frontier Postdoctoral Fellowship.

## Additional information

### Funding

| Funder | Author |
| --- | --- |
| Howard Hughes Medical Institute | Kang Shen |
| Human Frontier Science Program | Jing Yan |

The funders had no role in study design, data collection and interpretation, or the decision to submit the work for publication.

### Author contributions

JY, Conception and design, Acquisition of data, Analysis and interpretation of data, Drafting or revising the article; DLC, Performed experiments, Contributed unpublished essential data or reagents; ST, KK, TY, SH, Performed and analyzed the in vitro kinesin-1-MTs bundling assay, Acquisition of data, Analysis and interpretation of data; KS, Conception and design, Drafting or revising the article

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
