## [Decision Letter]

Thank you for choosing to send your work entitled “Kinesin-1 regulates Microtubule Polarity in the Dendrite” for consideration at *eLife*. Your article has been evaluated by a Senior editor and 3 reviewers, one of whom is a member of *eLife's* Board of Reviewing Editors.

The Reviewing editor and the other reviewers discussed their comments before we reached this decision, and the Reviewing editor has assembled the following comments based on the reviewers' reports.

Overall, the reviewers found the study of potentially high significance since it tackles an important and unresolved problem in neurobiology, namely the molecular mechanisms polarizing neurons and, in particular, the role of microtubules and molecular motors in axon/dendrite specification. The discovery that kinesin-driven microtubule transport might participate to the unique microtubule orientation characterizing the dendrite in *C. elegans* neurons is potentially interesting and novel.

However, at this point, the reviewers raised a number of concerns regarding the interpretation of some of the results, the validity of some the quantitative analysis, and some general aspects of the data presentation. If you can address the following major comments with new experiments, improved quantitative analysis, and revised text, we would like to consider a revised version.

1. Interpretation of the data: the authors claim that the conversion of microtubule orientation (minus end distal into plus end distal) in the kinesin-1 mutant is due to the microtubule transport function of kinesin-1, which might selectively transport minus-end microtubules to the distal end of the dendrite. The reviewers propose that, based on the plus-end direction of kinesin-1 transport, an equally likely interpretation of the authors' data is that kinesin-1 may function to promote transport of plus-end microtubules out of the dendrite into the cell body. The authors conclude the reverse (that ‘…dynein moves minus-end out MT out of the axon, while kinesin-1 moves minus-end out MT into dendrites'). The authors should consider this alternative interpretation of their results that kinesin-1 moves plus-end distal MTs out of the dendrites by using their plus-end directed motor activity to transport plus-end MT towards the cell body, that is out of the dendrite In the same vein, their conclusion schema (Figure 6G) is a bit confusing in this regard: in most of the paper, the authors represent minus-end microtubules distal in dendrites and the circle represents the cell body. In Figure 6G the circle indicates the distal end of the dendrite. The authors should make sure this is not a mistake. They should try to make clearer the difference between ‘long minus end out' MTs and the fragment/transported plus-end out MTs (maybe using a different color). The authors must discuss how they envision that kinesin-1 can establish minus end-out microtubules. If this motor were to slide microtubules along each other in dendrites, all of the microtubules would eventually collapse into the cell body. Finally, can the authors comment on the localization pattern of kinesin-1 in DA9? Is it found in the dendrite as the model in Figure 6G predicts?

2. More experiments on *Unc-116*: the authors should be more accurate when they describe the *e2310* allele of *Unc-116* (kinesin-1). For example, they mention that this allele ‘…likely represents a partial loss of kinesin activity'. But then they mention the study from Patel and colleagues, suggesting that *e2310* mutation affects a residue at the ‘…C-terminal end of kinesin-1 adjacent to the MT binding motif'. This is unlikely to affect kinesin-1 motor activity as suggested earlier. The authors argue that the C-terminal domain of *unc-116* is likely to bind microtubules based on sequence similarity (but not identity) of the MT binding domain to homologs, but whether the *C. elegans* sequence binds to MTs is not known. The major model presented by the authors will be strengthened by these data if the resources are available to perform this work. Finally, does *unc-116*-MT rescue the dendrite elongation defect of *unc-116* mutants? This experiment is relevant as the authors indicate in the Discussion that MT polarity appears to affect the length of the dendrite, when this could in principle be due to organelle trafficking roles.

3. More quantification: while the data are generally high quality, certain arguments would be strengthened by additional quantification, for example quantification is lacking from main Figures 1, 4, and 5.

4. One reviewer felt the authors were potentially misleading in the title and abstract by not indicating that they have investigated this in worms, rather than in mammals. The authors should improve their Introduction and how they formulate the question tackled. At the end of the first paragraph of the introduction, the authors state “dendritic MTs exhibit a mixed orientation with minus-ends predominantly facing the distal dendrite…”. This is not correct: it has been established that in mammalian systems, microtubules are of mixed polarity in the proximal region of the dendrite but are predominantly plus end-out in the distal dendrite (Baas 1988; Stepanova 2003; Kollins 2009; etc.). This has been very nicely reviewed in Kapitein and Hoogenraad, Mol Cell Neurosci 2011. The authors should improve their Discussion and quoting of the literature. For example, there are several more recent reviews with more relevant information than the ones cited (Hirokarwa 1998; [45]). More significantly, kinesin-2 has been shown to be required for minus end-out microtubule orientation in flies (Mattie et al Curr Biol 2010). The authors should cite this paper and discuss the relevance of their work to this paper.

5. It seems that dendritic outgrowth of the DA9 neuron occurs during post-embryonic development, suggesting that microtubule polarity of the dendrite is set up at L1. One reviewer feels it would be appropriate to demonstrate microtubule polarity via (+) and (-) end markers and the establishment of longer dendrites at L1 through adult stages to further support their hypothesis that in *unc-116* animals the dendrite takes on a longer, more axonal like configuration with reversed MT polarity.

[Editors'note: during revisions the editors made the following suggestion.]

The reviewers agreed that it would be beneficial to re-include the data regarding the dendrite elongation phenotype that you removed during the first revisions:

“[We] raised the issue of the basis for the dendrite elongation phenotype, which the authors have sorted out, but [we] felt that entirely removing the dendrite phenotype is not the best solution in the revision since the paper now refers to a lack of all other morphological defects (dendrite axon directionality and axon growth), but awkwardly omits dendrite length. The basis of this phenotype appears complex, yet [we] feel the authors should find a way to include at least the basic observation.”

---

## [Author Response]

We have followed the suggestions of the reviewers and performed additional experiments. The most constructive suggestion from the review was that our genetic data could be interpreted by two different models. One model is the one that we had originally presented: Kinesin-1 shuffles minus-end-out MTs into the dendrite. The alternative model is that Kinesin-1 shuffles plus-end-out MTs out of dendrite into the soma. We have performed two experiments to distinguish these two models. The new results favor the alternative model proposed by the reviewers. Accordingly, we have changed the manuscript to reflect these results. We have also included new in vitro data that directly support our model.

Since the in vitro experiments we added were not requested by the reviewers, we will describe them first before providing a direct reply to all the questions.

The genetic results we showed were consistent with the model that UNC-116/Kinesin-1 establishes the minus-end-out MT polarity in dendrites. Through structure-function analysis, we propose that kinesin-1 potentially binds to MTs with its C-terminal tail domain and slides anti-parallel MTs against each other. To directly test if Kinesin-1 can crosslink anti-parallel MTs, we collaborated with Dr. Hirotsune's laboratory, who performed in vitro MT cross-linking assays with the kinesin-1 complex. Consistent with previous literature, they found that kinesin-1 could cross-link MTs. They further used cryo-EM to determine the polarity of the cross-linked MTs. They found that the majority of MTs are of anti-parallel polarity (Figure 5). While these experiments were performed with vertebrate proteins, we feel that it provided valuable support for our model. We therefore included the data in the revised manuscript.

We want to be upfront about the low case number for the EM reconstruction of the MTs in this analysis. Since the assay is incredibly time consuming, we could only complete the reconstruction of 13 pairs of MTs, out of which 9 pairs are anti-parallel and 4 pairs are parallel. While the anti-parallel MTs accounts for the majority of the cases, it does not reach statistical significance based on these numbers. We are open to not including these data if the reviewers and editors advise us to do so.

Below is the point-by-pointreply to the questions raised by the reviewers.

*1) Interpretation of the data*.

We would like to thank the reviewers for coming up with this alternative model. Our original model is that “Kinesin-1 slides minus-end-out microtubules (MTs) into dendrites”. The alternative model states that “Kinesin-1 slides plus-end-out microtubules out of dendrites”.

The results we presented in the original manuscript are consistent with both models. Therefore, we performed two experiments to distinguish the two models. We reasoned that if the original model is correct, in *unc-116* mutants, the net number of MTs should decrease compared to the wild-type animals. On the contrary, the alternative model would predict that in *unc-116* mutants, the net number of MTs should increase. We used two methods to measure the relative concentration of MTs in dendrites.

First, we used a GFP::tubulin (GFP::TBA-1) transgene to assess the overall level of MTs. Since the absolute fluorescence intensity is affected by the expression level of the transgenes, we used the ratio between the dendrite and axon fluorescence as a measure for the relative enrichment of MTs in the dendrite vs. axon. We found that *unc-116* mutants show significantly higher MT abundance compared to the wild type (Figure 6E, error bars represent standard deviation).

Second, we analyzed the frequency of the EBP-2::GFP comets as a measure of the number of dynamic MTs in DA9 dendrites. Interestingly, we again found that *unc-116* mutant dendrites have three times higher frequency of EBP-2::GFP comets compared to the wild-type controls. No difference was found in the axons between mutant and wild type.

Together, both experiments support the alternative model and suggest that Kinesin-1 regulates the minus-end-out MT polarity in the dendrite by specifically transporting plus-end-out MTs out of the dendrite. In the *unc-116* mutants, plus-end-out MTs accumulate in the dendrite and lead to the erosion of polarity.

Regarding the conclusion schema (Figure 6G), the symbol was indeed a mistake. We apologize for that and we have corrected it. We have also distinguished the longer MTs and shorter transporting MTs with two different colors as the reviewers suggested. We envision that the kinesin-1 slides plus-end-out MTs out of the dendrite to create the dominant polarity of minus-end-out MTs in the DA9 dendrite. We envision that this function must be regulated. From the EBP-2::GFP experiments, it is clear that a small percentage of MTs are plus-end-out even in the wild type dendrites.

We have also created UNC-116::GFP to examine the localization of this motor in DA9. Similar to the UNC-104::GFP construct, we found that the vast majority of fluorescence is found at the tip of the axon, again supporting the plus-end-out MT polarity in the axon. We do not find a high level of UNC-116::GFP in the dendrite as expected. However, these experiments are not perfect since they detect transgenically expressed UNC-116. In this case, even antibody staining will likely not be useful, as it will lose the cellular specificity attained by the DA9 dendrite being tightly associated with the ventral nerve cord. It is therefore impossible to differentiate the potential signal from DA9 and those from the many other neurites. One related note is that kinesin-1 was found in the vertebrate dendrites (Nakata and Hirokawa, 2003).

*2) More experiments on* Unc-116.

We have sequenced the *e2310* allele and confirmed the previously reported TC5 transposon insertion. The transposon was inserted into the C-terminal coding region of *unc-116*, which results in a premature stop codon after residue 692, which truncates the potential MT-binding domain and the very C-terminal IAK motif of kinesin-1/UNC-116. The IAK motif was shown to be important for regulating the motor activity of kinesin-1 (Wong et al., 2009). So we anticipate that *e2310* allele will cause impaired motor activity.

In order to test whether the putative MT-binding domain of UNC-116 is able to bind MTs, we expressed GFP-tagged UNC-116(aa521-863) in the HEK 293T cells and co-stained with MTs. As shown in the following figures, GFP-UNC-116(aa 521-863) co-localized with MTs (Author response image 1). Similar experiments were used to demonstrate that the vertebrate kinesin-1 counterpart can directly interact with MTs (Navone et al., 1992). Importantly, a mutant form of this domain that includes three R to A mutations showed no colocalization with MT markers. The same mutations when introduced in the vertebrate Kinesin-1 domains also diminish the interaction between MT and the C-terminal MT binding motif (Seeger and Rice, 2010). Together, these results further support our model.Author response image 1.

We want to thank the reviewers for raising questions about the dendrite length phenotype in *unc-116*. In our original submission, we showed that in *unc-116* mutants, the DA9 dendrite exhibits axon-like MT polarity and is also significantly longer than the wild type dendrite. The reviewers asked us to test whether the increased length is caused by the change of MT polarity. Through structure-function dissection of UNC-116, we identified a mutant form of UNC-116 which is defective in the putative novel MT sliding function, yet appears to maintain other vesicular trafficking functions as measured by the localization of mitochondria (Figure 6B and Figure 6—figure supplement 1). When expressed in DA9, this mutant UNC-116 (UNC-116 _MT_) failed to rescue the MT polarity phenotype, but fully rescued the dendrite length phenotype, indicating the longer dendrite is not a consequence of MT polarity change. Consistent with this notion, we examined the dendrite length of another mutant *wy774*, which we have isolated from our forward genetic screen. This mutant exhibits similar MT polarity phenotype as the *unc-116* mutants but is mapped to a different genomic locus (hence not another allele of *unc-116*). Interestingly, this mutant shows normal dendrite length, further suggesting that longer dendrite in the *unc-116* mutant is not a consequence of MT polarity change but likely due to other functions of UNC-116/kinesin-1. In the light of these new results, we have removed the data and discussion on dendrite length in the revised manuscript.

*3) More quantification*.

We have included quantifications in Figures 1, 4 (now Figure 3) and 5 (now Figure 4).

*4) One reviewer felt the authors were potentially misleading in the title and abstract by not indicating that they have investigated this in worms, rather than in mammals […]*.

We have clarified the organism in the title and abstract. We have modified our introduction on the dendrite MT polarity, cited a number of recent reviews, and included discussion on kinesin-2.

*5) [It] would be appropriate to demonstrate microtubule polarity* via *(+) and (-) end markers and the establishment of longer dendrites at L1 through adult stages […].*

We agree with the reviewers that this is an important issue. Using +end and –end markers as suggested by the reviewers, we examined the DA9 dendrite polarity at different developing stages in both wild type and *unc-116* mutants. The results are summarized below and shown in Author response image 2.Author response image 2.

1) In wild type L1s (10 hour after hatching) animals, (-) end marker KLP-16::GFP is localized to the dendrite, but does not show the dramatic enrichment at the tip of the dendrite found in adults. At the same stage, (+) end marker UNC-104::GFP avoids the dendrite and is dramatically enriched at the tip of the DA9 axon. These results suggest that in wild type young animals, the dendrite likely already contains mix-polarized MTs but the minus-end-out MTs are not as dominant as in adult dendrites. We looked hard for any evidence that would suggest that very young dendrites bear axonal-like polarity, a phenotype that was observed in the stage 2 neurons in dissociated vertebrate cultures. However, we did not see any UNC-104::GFP fluorescence in the dendrites of wild-type animals at any developmental stages. We therefore conclude that as soon as the dendrite grows out, it has adopted a mixed MT polarity.

2) at 20 hours after hatching, wild type DA9 dendrite showed dramatic enrichment of (-) end marker KLP-16::GFP at the tip of the dendrite, suggesting that the minus-end-out MTs have become dominating species by this time point.

3) at 10 hour after hatching, *unc-116* mutant dendrites showed the same distribution of (-) end marker and (+) end marker as in the wild type. In other words, *unc-116* mutant did not show a phenotype at this very early stage of dendrite development.

4) at 20 hours after hatching, the *unc-116* mutant dendrite completely fails to enrich the (-) end marker KLP-16::GFP at its tip; instead, it enriches the (+) end marker UNC-104::GFP at its tip, suggesting a polarity reversal starting at this stage.

Taken together, these results suggest that the dendrite starts out as a process with a mixed MT polarity (certainly not an axon-like uniformly plus-end-out, but also not as minus-end-out as the mature dendrite). With time, MTs becomes more dramatically polarized with minus-end-out being the dominating species.UNC-116/Kinesin-1 likely functions in the maturation stage to establish and maintain the polarity. These data are consistent with our MT gliding model.

*[Editors' note: during revisions the editors made the following suggestion: The reviewers agreed that it would be beneficial to re-include the data regarding the dendrite elongation phenotype that you removed during the first revisions.*]

We have followed your suggestions and put back the DA9 dendrite length elongation phenotype of *unc-116* animals in the manuscript as a supplement to Figure 3.

**References**

Nakata, T, & Hirokawa, N. (2003). Microtubules provide directional cues for polarized axonal transport through interaction with kinesin motor head. *J Cell Biol, 162*, 1045–55. doi: 10.1083/jcb.200302175

Navone, F, Niclas, J, Hom-Booher, N, Sparks, L, Bernstein, HD, McCaffrey, G, et al. (1992). Cloning and expression of a human kinesin heavy chain gene: interaction of the COOH-terminal domain with cytoplasmic microtubules in transfected CV-1 cells. *J Cell Biol, 117*, 1263–75. doi: 10.1083/jcb.117.6.1263.

Seeger, MA, & Rice, SE. (2010). Microtubule-associated protein-like binding of the kinesin-1 tail to microtubules. *J Biol Chem, 285*, 8155–62. doi: 10.1074/jbc.M109.068247

Wong, YL, Dietrich, KA, Naber, N, Cooke, R, & Rice, SE. (2009). The Kinesin-1 tail conformationally restricts the nucleotide pocket. *Biophys J, 96*, 2799–807. doi: 10.1016/j.bpj.2008.11.069.